# Evaluation of Three Marine Algae on Degradability, In Vitro Gas Production, and CH$_4$ and CO$_2$ Emissions by Ruminants

Héctor Aarón Lee-Rangel [1,*] , José Alejandro Roque-Jiménez [1,2,*] , Rubén Oswaldo Cifuentes-López [1],
Gregorio Álvarez-Fuentes [1], Adriana De la Cruz-Gómez [1], José Antonio Martínez-García [2],
Jaime Iván Arévalo-Villalobos [1] and Alfonso Juventino Chay-Canul [3]

[1]   Centro de Biociencias, Instituto de Investigaciones en Zonas Desérticas, Facultad de Agronomía y Veterinaria,
      Universidad Autónoma de San Luis Potosí, km 14.5 Carr. San Luis Potosí—Matehuala,
      San Luis Potosí 78321, Mexico
[2]   Departamento de Producción Agrícola y Animal, Universidad Autónoma Metropolitana-Xochimilco,
      Mexico City 04960, Mexico
[3]   División Académica de Ciencias Agropecuarias, Universidad Juárez Autónoma de Tabasco, Carretera
      Villahermosa-Teapa, km 25, R/A. La Huasteca 2a Sección, Villahermosa 86280, Mexico
*     Correspondence: hector.lee@uaslp.mx (H.A.L.-R.); jroque@correo.xoc.uam.mx (J.A.R.-J.);
      Tel.: +52-44-4293-7922 (J.A.R.-J.)

**Abstract:** Livestock production systems are responsible for producing ~18% of the total anthropogenic greenhouse gas (GHG) emissions. Numerous alternatives, such as feed additives or supplements, have been proposed to meet these challenges. Marine algae have been proposed for gas reduction in ruminants; nevertheless, there are still very few experimental reports. Thus, the objective of the current study was to identify the volatile organic compounds (VOCs) in three marine algae—Kelp (*Macrocystis pyrifera*), Ulva (*Ulva* spp.), and Silk (*Mazzaella* spp.)—and to test their in vitro ruminal fermentation characteristics, gas profiles, and ability to mitigate biogas production. The evaluation of the VOCs in marine algae was performed using a flash gas chromatography electronic nose (FGC-E-Nose). The in vitro study was elaborated through in vitro incubation and gas production. The data obtained were analyzed using a completely randomized design. In total, forty-three volatile compounds were identified for Kelp algae, thirty-eight were identified for Ulva algae, and thirty-six were identified for Silk algae; the compounds were from different chemical families and included aromas, alcohols, aldehydes, phenolics, carboxylic acids, esters, and nutraceutical properties. Dry matter degradability was significantly ($p < 0.05$) affected by the algae type. The cumulative ruminal gas production was different ($p < 0.05$) between treatments. Kelp algae presented a major (V; $p < 0.05$) volume of gas produced compared to the other algae. Lag time (l; $p < 0.05$) was increased by Kelp alga; however, there were no differences (p>0.05) between the Silk and Ulva algae. The gas production rate was higher (s; $p < 0.05$) for Silk algae compared to the others. Ulva and Silk algae demonstrated a significant ($p < 0.05$) decrease in carbon dioxide emissions. Nevertheless, Kelp algae reduced the proportional methane (CH$_4$) production ($p < 0.05$) after 48 h of incubation, with the lowest emission rate of 47.73%. In conclusion, algae have numerous bio compounds that provide some properties for use in ruminant diets as additives to reduce methane and carbon dioxide emissions.

**Keywords:** gas in vitro; algae; ruminant; methane; volatile organic compounds

## 1. Introduction

The demand for food of animal origin is expected to increase by 2050 due to the increased population in urban areas and the direct relationship with increases in income in some strata of society [1]. However, what will have the most significant impact will be the effects of climate change (droughts or floods), a consequence of the increase in greenhouse gas (GHG) concentrations, such as carbon dioxide (CO$_2$), methane (CH$_4$), and nitrous oxide (N$_2$O), among other atmospheric gases [2]. Steinfeld et al. [3] mentioned that the livestock

sector emits 7.1 Gt of $CO_2$, equivalent to approximately 18% of global anthropogenic emissions. Methane gas is the product of anaerobic fermentation in ruminant species and, together with carbon dioxide and nitrous oxide, represents the main GHG expelled by ruminants; methane emissions have a warming potential that is 25 times higher than that of carbon dioxide; and it has a half-life of 9 to 15 years in the atmosphere [4], a relatively short period compared to other GHGs. Methane is a product that results from the fermentation of carbohydrates in ingested feed; during this fermentation process, the methanogenic archaea present in the rumen use $CO_2$ and $H_2$ as a substrate to form $CH_4$ and to thus reduce the accumulation of $H_2$ in the rumen, avoiding digestive and metabolic problems in the animal [5]. Besides the effects on the environment, methane represents a loss of energy from the feed consumed by the animal. Therefore, it is an expression of inefficiency in production that is variable and can represent between 5 and 18 % of the total energy consumed by ruminants [6]. Currently, most studies on livestock and climate change in Latin America have focused on quantifying $CH_4$ emission volumes, determining emission factors, and calculating national inventories; few studies focus on developing mitigation strategies [5].

Macroalgae are complex and multicellular organisms found in various habitats and that have a long history of being used as livestock ruminant feed. [7]. Makkar et al. [7] identified several positive attributes of macroalgae concerning contributing nutrients, such as protein and the energy metabolism of animals of livestock interest. An important point is the bioactive compounds that could favor the production and animal health status. Currently, some studies on the effects of algae as a supplement in ruminants are under development; the authors of [8] mentioned that algae supplements improved the mineral status of animals, especially iodine and selenium. Some in vitro studies have shown that both red macroalgae and brown macroalgae can reduce CH4 production, with freshwater green algae having the slightest effect. A problem in these studies is the dose used and the fact that there are no reports on the bioactive compounds of the algae used. Li et al. [9] demonstrated that using *Asparagopsis taxiformis* in sheep can reduce methane emissions by 80%. Although derived from this study, an overestimation of both doses and CH4 emission calculation was demonstrated. Because of the above, it is necessary to determine the bioactive compounds in the algae to be used as a supplement in ruminants and to carry out determinations and effects on ruminal fermentation. Thus, the objective of the current study was to identify the volatile aromatic bio compounds of three types of marine algae—Kelp (*Macrocystis pyrifera*), Ulva (*Ulva* spp.), and Silk (*Mazzaella* spp.)—and to test their in vitro ruminal fermentation characteristics, gas profiles, and ability to mitigate biogas production.

## 2. Materials and Methods

### 2.1. Algae Harvesting

The three marine algae were harvested from an agro-fisheries site in the Sea of Cortés (Gulf of California, Lat Long, 31°38′23.4″N 114°42′49.9″W). Later, algae were processed by the Bajakelp® enterprise in La Bocana de Santo Tomás, Baja California, México.

### 2.2. Evaluations of Volatile Organic Compounds (VOCs) in Kelp, Silk, and Ulva by Flash Gas Chromatography Electronic Nose

A flash gas chromatography electronic nose (FGC-E-Nose) model, Heracles II, equipped with an automatic injection unit HS100 (AlphaMOS®, Tolouse, France) was used to detect the VOCs from the three marine algae: Kelp (*Macrocystis pyrifera*), Ulva (*Ulva* spp.), and Silk (*Mazzaella* spp.). The Heracles II was equipped with two columns working in parallel mode: a non-polar column (DB-5: 5% phenyl/95% dimethylpolysiloxane) and DB-1701 (14% cyanopropylphnyl/86% dimethylpoly-siloxane). The injector was maintained at a constant temperature of 200 °C.

The algae samples were placed in 20 mL magnetically sealed vials with a plug and without any treatment or extraction solvent. The vials were placed in the Heracles II

auto-sampler, which was placed in a shaker oven and shaken at 500 rpm for 900 s at 40 °C. Next, a 1 mL sample was taken from the headspace in the electronic nose. Samples were analyzed in triplicate. A single chromatogram was created by joining two columns of overlapping chromatograms, helping to reduce identification errors. Identifications were made using the Kovats index [10]. The GC subjected the samples to a temperature program, separating the volatile organic compounds and maintaining a constant hydrogen flow of 1 mL/min. Then, the samples were brought to a temperature of 50 °C for 30 s before increasing it by 10 °C/s until it reached 280 °C. Separate species were detected by the electronic nose software using multi-variable statistical analysis (Alpha Soft®by Alpha MOS®, Tolouse, France).

Chemometrics: In this study, the first explorative step was carried out using peak areas that were automatically calculated using Alpha Soft®software, which uses raw data from the abundance of metabolites to construct a multivariate model using Principal Component Analysis (PCA). PCA uses an orthogonal transformation to convert a set of observations of the different compounds of possibly correlated variables into values of linearly uncorrelated variables. This analysis guaranteed independence if the group of data was jointly normally distributed. PCA is a chemo-metric procedure that rotates the original space to another one, and its vector results are the principal components (PC), which are oriented along directions containing the maximum explained variance [10].

### 2.3. Chemical Composition Analysis and In Vitro Fermentation Kinetics

Dry matter (DM), crude protein (PC), ether extract (EE), and ash were determined in the samples according to the methodology described in the AOAC [11]. Neutral detergent fiber (NDF) and acid (ADF) were quantified according to Van Soest et al. [12] methodology.

Experimental samples of three marine algae—Kelp (*Macrocystis pyrifera*), Ulva (*Ulva* spp.), and Silk (*Mazzaella* spp.)—were used as substrates for in vitro incubation. The samples were dried at 65 °C for 24 h. Once dried, the samples were milled through a sieve with a mesh of 1 mm of diameter. For incubation, 0.5 g of the algae sample was added to 90 mL of culture medium.

The culture media used were similar to the one described by Cobos and Yokoyama [13]. Briefly, they contained mineral solution I (6 g of $K_2HPO_4$ in 1 L of distilled water), mineral solution II (6 g of $K_2HPO_4$; 6 g $(NH_4)_2SO_4$; 12 g NaCl; 2.45 g $MgSO_4$; and 1.6 g $CaClH_2O$ in 1 L of distilled water), 18% sodium carbonate buffer solution, reduced cysteine solution (2.5 g of L-cysteine in 15 mL of NaOH (2N)), 2.5 g of $Na_2S$, and 0.1 mL of resazurin (1%).

Ruminal fluid was obtained from two Rambouillet ewes with an average weight of 75 kg and fed a diet containing 50% corn silage and 50% alfalfa hay using a vacuum pump via an esophageal tube before feeding. The sampling procedures in the ewes were reviewed and approved by the Committee for the Ethical Use of Animals in Experiments of the Universidad Autonoma Metropolitana and Universidad Autonoma de San Luis Potosi (BP-PA-20210513092803518-1069570). The ruminal fluid pH was measured immediately with a pH meter (Benchtop Cole Parmer 05669-20, Vernon Hills, IL, USA). The average pH levels were from 6.9 to 7.1. After the ruminal fluid was obtained, it was filtered through gauze and was kept at 39 °C until later use. The ruminal fluid was added to the culture media at a final concentration of 10% [14]. A total of 0.5 g of each of the three marine algae, Kelp (*Macrocystis pyrifera*), Ulva (*Ulva* spp.), and Silk (*Mazzaella* spp.), was added to a 120 mL vial with 90 mL of the inoculum, and vials with only 90 mL of inoculum were used as controls. The vials were incubated in a water bath at 39 °C.

Gas production was measured following the procedure described by Getachew et al. [15]. The gas produced was calculated at different times using a hypodermic needle connected to a 0–1 kg/cm$^2$ gauge inserted in the vial plug. The pressure was measured at 0.5, 1, 2, 4, 6, 8, 10, 12, 16, 24, 36, and 48 h after incubation. The pressure (kg/cm$^2$) was converted to volume and accumulated gas production using the model proposed by Menke and Steingass [14]:

$$Y = v / (1 + \exp (2\text{-}4 \times s \times (t \text{ - } L))) \tag{1}$$

where

    Y = total gas produced (mL);
    v = volume ($mL/g^{-1}$);
    s = gas production rate ($mL/g^{-1}$);
    t = time (h):
    L= lag phase (h).

At the end of incubation (96 h), the contents of each serum bottle were filtered using sintered glass crucibles (coarse porosity no. 1, 100 to 160 m pore size, Pyrex, Stone, UK) under vacuum. Fermentation residues were dried at 105 °C overnight to estimate the DM disappearance.

Applying the model proposed by Menke and Steingass [14], the metabolizable energy (ME, MJ/kg of DM) was calculated as follows:

$$\text{ME, MJ/kg of DM} = 2.20 + 0.1357GP24 + 0.0057CP + 0.0002859EE2 \tag{2}$$

where

    GP24 = amount of gas produced after 24 h of incubation;
    CP = crude protein of diets (% DM);
    EE = ether extract of diets (%DM).

The concentration of short-chain fatty acids (mmol SCFA) was calculated according to the equation proposed by Getachew et al. [15]:

$$\text{mmol SCFA} = -0.00425 + 0.0222 \text{ (mL gas at 24 h)}$$

### 2.4. Methane (CH$_4$) and Carbon Dioxide (CO$_2$) Production

The volume and production of $CO_2$ after 4, 8, 12, 24, 36, 48, and 72 h of fermentation were measured volumetrically, and the methane and minor gases were estimated according to the differences in these values (Singh and Mohini, 1999). Thirty milliliters of reduced mineral solution and rumen fluid (2:1) were added into amber flasks (60 mL) containing 0.25 g of each diet. Gas production was measured at 0, 6, 12, and 24 h with a 150 mL glass syringe to determine the production of $CO_2$ and minor gases. Sequential selection of each tube outflow to an infrared gas analyzer (IRGA) (Li-6262, LiCor, Lincoln, NE, USA) allowed us to monitor $CO_2$ mixing ratios corrected for the effects of water vapor in real-time. The IRGA was calibrated with secondary standards traceable to NOAA CMDL standards before and after each measurement. The span and zero drifts were less than 1 ppm. The analytical accuracy was traceable to NOAA CMDL standards.

### 2.5. Statistical Analysis

The data obtained were analyzed using a completely randomized design [16]. The PROC NLIN BEST method (SAS Institute, 2002, NC, USA) was performed to calculate the ruminal in vitro kinetics (volume of gas produced, gas production rate, and initial gas production). Data were analyzed using JMP7 software [16]. A *p*-value of 0.05 was selected as the significance level.

## 3. Results

### 3.1. The VOCs Present in Kelp, Silk, and Ulva Algae Determined by FGC- E-Nose

The Kovats retention index (KRI) was used to convert the retention times into system-independent constants. The KRI database allowed the identification of 43 relevant VOCs in Kelp algae, 38 in Ulva algae, and 36 in Silk algae (Table 1), with the different VOCs including aromas, alcohols, aldehydes, and phenolics, some with nutraceutical properties.

**Table 1.** Tentative identification of VOCs in Kelp, Ulva, and Silk algae by the FGC- E-Nose profile.

| Kelp | | | Ulva | | | Silk | | |
|---|---|---|---|---|---|---|---|---|
| RT min | Compound | Relevance Index | RT min | Compound | Relevance Index | RT min | Compound | Relevance Index |
| 12.69 | Butane | 44.45 | 12.7 | Butane | 51.14 | 15.11 | Butane | 56.94 |
| 13.02 | Butane | 50.93 | 13.02 | Butane | 56.94 | 17.67 | propanon-2-one | 79.55 |
| 14.57 | Trimethylamine | 44.82 | 14.57 | Trimethylamine | 49.06 | 18.68 | Diethyl ether | 57.98 |
| 15.53 | 2-Methylbutane | 39.77 | 15.53 | 2-Methylbutane | 43.72 | 19.78 | 1-Propanol | 83.79 |
| 17.01 | Diethyl ether | 57.97 | 17.01 | Diethyl ether | 57.98 | 20.85 | Carbon disulfide | 73.05 |
| 18.15 | 1,1-Dichloroethene | 55.69 | 19 | Propanon-2-one | 79.55 | 22.38 | butan-2-one | 80.93 |
| 19.04 | propanon-2-one | 57.39 | 20.26 | Diisopropyl ether | 70.53 | 24.21 | Trichloroethane | 39.48 |
| 20.27 | Ethene, 1,2-dichloro-, (E) - | 71.32 | 21.67 | 2-methylfuran | 79.10 | 25.13 | 2-Octenal, (E) | 68.34 |
| 21.69 | 1-Propanol | 62.40 | 22.5 | 1-Propanol | 83.79 | 27.17 | Acetoin | 77.74 |
| 22.55 | Carbon disulfide | 72.87 | 23.33 | butan-2-one | 80.93 | 28.3 | Methyl butanoate | 86.03 |
| 23.35 | 2-butanol | 85.82 | 24.16 | 2-butanol | 60.47 | 30.54 | pentanol | 81.53 |
| 24.17 | 2-butanol | 58.07 | 26.91 | Methyl butanoate | 86.03 | 32.39 | Octane | 70.26 |
| 25.11 | 1,2-Dichloroethane | 86.97 | 27.62 | 2,3-Pentanedione | 71.64 | 35.01 | E-2-Hexen-1-ol | 93.20 |
| 26.93 | 1,2-Dichloropropane | 80.11 | 30.04 | Acetoin | 77.74 | 37.09 | pentanoic acid | 82.72 |
| 27.66 | pental-2-ol | 76.88 | 30.82 | pentanol | 81.53 | 38.21 | Methyl hexanoate | 87.25 |
| 30.11 | Acetoin | 57.89 | 31.52 | Ethyl isovalerate | 83.32 | 39.86 | 1-Heptanol | 86.44 |
| 30.83 | pentanol | 72.40 | 33.89 | ethyl pentanoate | 73.53 | 40.63 | 2,3-Octanedione | 91.82 |
| 31.57 | Ethyl isovalerate | 86.62 | 34.44 | E-2-Hexen-1-ol | 93.20 | 41.46 | 2-Ethyl-3-methylpyrazine | 86.08 |
| 35.43 | (-) – beta,-Pinene | 84.61 | 35.38 | sabinene | 73.92 | 42.56 | acetilpyrazine | 77.41 |
| 36.03 | alfa-Phellandrene | 89.23 | 35.98 | alpha-Pheladrene | 83.34 | 43.43 | (Z)-2-octenal | 83.08 |
| 37.86 | Putrescine | 91.25 | 37.79 | 2,3-Octanedione | 91.82 | 44.11 | 1,2-Cyclopentanedione | 85.73 |
| 39.08 | Benzene, 1,2-dichloro | 80.26 | 38.59 | Undecane | 92.20 | 45.23 | Undecane | 92.20 |
| 39.76 | 2-Isopropyl-3-methoxypyrazine | 85.22 | 39.01 | terpinolene | 76.18 | 45.72 | p-menthatriene | 85.28 |

**Table 1.** *Cont.*

| Kelp | | | Ulva | | | Silk | | |
|---|---|---|---|---|---|---|---|---|
| RT min | Compound | Relevance Index | RT min | Compound | Relevance Index | RT min | Compound | Relevance Index |
| 40.45 | p-menthatriene | 82.41 | 39.69 | acetilpyrazine | 77.41 | 46.19 | Limonene oxide | 79.92 |
| 41.36 | ethenyl-dimethylpyrazine | 85.79 | 40.41 | p-menthatriene | 85.28 | 47.73 | 2,3-Diethyl-5-methylpyrazine | 42.15 |
| 42.65 | ethyl 3-(methylthio)propanoate | 90.90 | 41.27 | ethenyl-dimethylpyrazine | 82.34 | 49.74 | Decanal | 88.12 |
| 43.19 | 1,2-Cyclopentanedione, 3,4 … | 74.89 | 42.58 | Limonene oxide | 79.92 | 50.5 | 2,6-Dichlorophenol | 87.04 |
| 45.03 | Triethyl phosphate | 77.13 | 43.16 | 1-2-Cyclopentanedione, 3,4 | 85.73 | 52.32 | Anethole | 81.06 |
| 45.85 | p-Cresol | 79.15 | 44.96 | Decanal | 88.12 | 53.17 | ndecane-2-one | 72.60 |
| 47.45 | N,N-dimethylacetamide | 65.62 | 45.78 | Methyl salicylate | 79.22 | 56.09 | trans-2-Undecenal | 79.06 |
| 48.78 | Tetradecane | 87.87 | 47.38 | Nerol | 78.86 | 57.22 | Tetradecane | 80.74 |
| 49.85 | 5-ethyl-3-hydroxy-4-methyl-2 … | 95.20 | 48.72 | 2,6-Dichlorophenol | 87.04 | 59.49 | Carbamothioic acid, butyleth | 33.39 |
| 51.62 | trans-2-Undecenal | 81.81 | 49.77 | 2,4-Decadienal, (E;Z) | 72.25 | 60.94 | beta-Himachalene | 41.58 |
| 54.41 | beta-Himachalene | 38.97 | 51.57 | trans-2-Undecenal | 79.06 | 63.91 | Rheosmin | 30.65 |
| 55.57 | beta-ionone | 35.18 | 54.25 | gamma-nonalactona | 48.98 | 74.38 | 1,4-Naphthalenedione | 28.00 |
| 56.88 | Mevinphos | 75.53 | 55.51 | beta-ionone | 34.85 | | | |
| 57.56 | wine lactone | 87.26 | 67.13 | 1,4-Naphtalenedione, 2,3 … | 28.00 | | | |
| 58.44 | delta-decalactone | 84.64 | 70.3 | Chlorothalonil | 24.51 | | | |
| 60.53 | Tebuthiuron | 76.31 | | | | | | |
| 61.95 | Acetamide, 2-chloro-N | 50.55 | | | | | | |
| 63.44 | 4(4-hydroxy-3-methoxypheny | 54.14 | | | | | | |
| 64.44 | 3-oxo-alpha-ionone | 29.49 | | | | | | |
| 67.04 | 1,4-Naphtalendione, 2,3 | 25.03 | | | | | | |

The data obtained from the samples were analyzed via PCA (Figure 1), and natural separation was observed within the study groups, indicating that the VOCs have different chemical prints. PC1 presents a percentage of explanation of 65.44 %, with PC2 reaching 23.13.2 % and PC3 reaching 7.43 %, resulting in a cumulative percentage of explanation of the variation of 96 %.

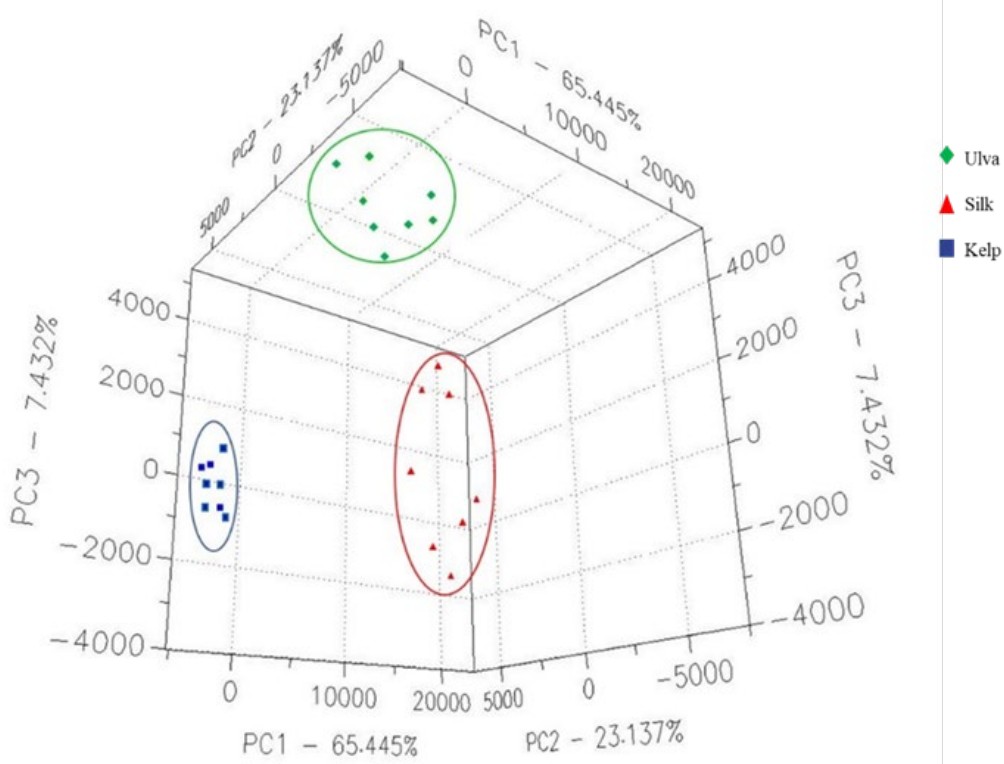

**Figure 1.** PCA model built with the electronic nose data related to the Kelp, Ulva, and Silk algae.

*3.2. Chemical Composition, In vitro Dry Matter Degradability (IVDMD), and In Vitro Fermentation Kinetics*

The chemical composition (Table 2) of the different algae showed numerical differences with crude protein (CP). Additionally, Ulva algae showed a lower concentration of ether extract (EE) than the other algae. Interestingly, Silk algae have little ether extract.

**Table 2.** Chemical composition (% DM basis) of the Ulva, Silk, and Kelp algae.

|  | Ulva | Silk | Kelp |
|---|---|---|---|
| DM, % | 94.5 | 96.1 | 99.3 |
| OM, % | 78.6 | 79.1 | 77.1 |
| CP, % | 6.6 | 10.5 | 10.4 |
| EE, % | 0.33 | 0.19 | 0.41 |
| NDF, % | 32.3 | 39.2 | 38.6 |
| ADF, % | 43.4 | 46.2 | 40.1 |
| Ash, % | 21.4 | 20.9 | 22.9 |

DM, dry matter; OM, organic matter; CP, crude protein; EE, ether extract; NDF, neutral detergent fiber; ADF, acid detergent fiber.

During the ruminal incubation of the algae, Kelp and Ulva presented similar ($p < 0.05$) degradability, but after 48 h, they differed from Silk algae. Significant ($p < 0.05$) differences were found among treatments after 72 h. Differences ($p < 0.05$) were observed in Silk compared to the other treatments from 0 to 72 h (Figure 2).

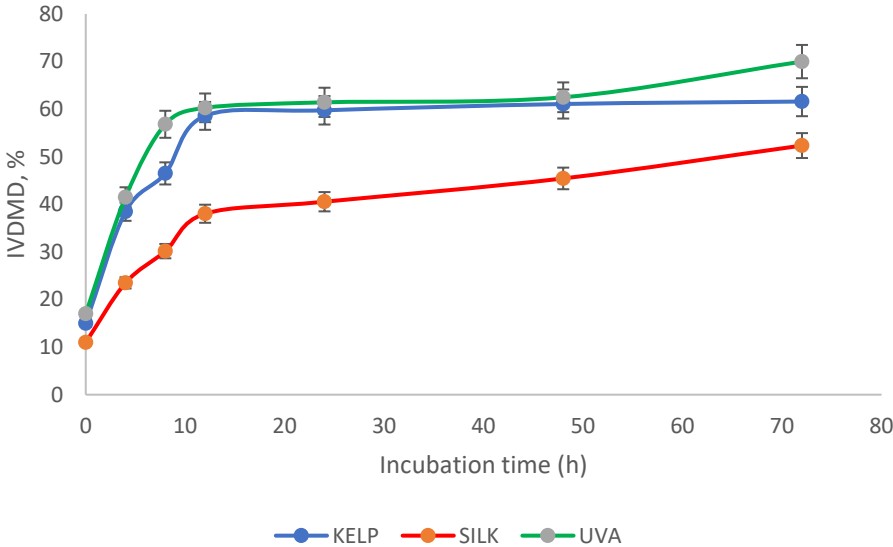

**Figure 2.** In vitro dry matter degradability of the Kelp, Silk, and Ulva algae after 72 h.

The volume of gas produced (V) presents a significant ($p < 0.05$) difference between treatments, with Kelp and Ulva algae (63.7 and 44.37), having a greater in vitro volume production compared to Silk alga (30.33). The lag phase was prolonged ($p < 0.05$) with Kelp algae. Silk algae present a higher ($p > 0.05$) gas production rate than the other treatments. The proportional production of $CH_4$ and $CO_2$ was affected ($p < 0.05$) by the type of algae used (Table 3).

**Table 3.** In vitro cumulative gas production and estimated kinetic parameter model for Kelp, Ulva, and Silk algae.

|  | **Kelp** | **Ulva** | **Silk** | **SEM** |
|---|---|---|---|---|
| The volume of gas produced (v), mL | 63.4[a] | 44.37[ab] | 30.33[c] | 12.9 |
| Production rate (s), $mL/g^{-1}$ | 0.017[b] | 0.015[b] | 0.025[a] | 0.001 |
| Lag time (L), h | 17.29[a] | 9.49[b] | 9.47[b] | 2.3 |
| % $CO_2$ at 48 h | 52.27[a] | 38.97[ab] | 28.38[c] | 4.81 |
| % $CH_4$ at 48 h | 47.73[c] | 61.03[b] | 71.62[a] | 9.78 |
| ME, MJ/kg of DM | 4.06 | 3.54 | 3.48 | 0.85 |
| mmol SCFA | 0.205 | 0.295 | 0.196 | 0.003 |

[a,b,c] Means within row with different superscripts differ ($p < 0.05$); SEM, standard error of the mean.

### 3.3. Ruminal $CO_2$ and $CH_4$ Production

Figure 3 depicts in vitro ruminal $CO_2$ and $CH_4$ emissions (mL/0.5 g incubated DM) resulting from the use of Kelp, Silk, or Ulva algae as ruminant feed. A significant difference ($p > 0.05$) was observed in $CO_2$ production (mL/0.5 g incubated DM) due to the use of different algae. The $CO_2$ production (mL/0.5 g degraded DM) was increased by Silk algae. In vitro ruminal $CH_4$ emissions (mL/0.5 g incubated DM) influenced due to the use of various algae types as a ruminant feed are shown in Figure 3. Kelp algae revealed CH4 emission mitigation (mL/0.5 g incubated DM). The $CH_4$ emissions (mL/0.5 g degraded DM) were affected ($p < 0.05$) after 24 and 48 h after the onset of algae use.

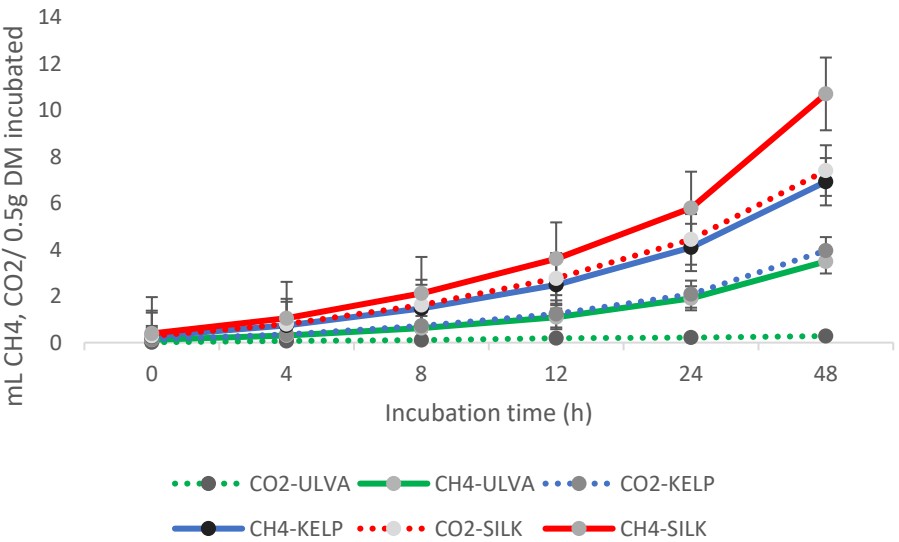

**Figure 3.** Effects of Kelp, Ulva, and Silk algae on ruminal $CO_2$ and $CH_4$ in vitro production (mL/0.5 g incubated DM).

## 4. Discussion

The analysis results obtained by FGC- E-Nose reveal a significant difference among the volatile compounds. We expected that the volatile profile would be different from that of terrestrial vegetable material. Aromatic phytochemical screening of Silk, Ulva, and Kelp algae showed that the seaweeds contain flavonoids, alkaloids, terpenes, and saponins [17]. Principal Component Analysis (PCA) is a functional tool that can discriminate based on chemical composition. Green, red, and brown macroalgae have been discriminated against based on distinct concentrations and types of fatty acids during evaluations tropical macroalgae [18].

Berneira et al. [19] mentioned that brown macroalgae present aldehydes, as does red seaweed; these are derived from the metabolization of polyunsaturated fatty acids, as well as through the amino acid pathway [20]. Carbonyl compounds such as ketones and aldehydes function as attractants or repellents in terrestrial plants, but in marine organisms, their function still is unknown [21]. Ketones, the octane compound found in Silk algae, is derived from the metabolization of fatty acids, carotenoids, and amino acids [22]. Additionally, they contribute to aroma due to their low odor thresholds and the beta-ionone present in Ulva and Kelp algae [20]. Berneria et al. [23] studied sub-Antarctic macroalgae and found that hydrocarbons were another representative group of VOCs, representing as much as 88.94% of the composition. It is important to note that the VOCs in macroalgae that act on the chemical signals (pheromones) are acyclic, unsaturated, and alicyclic (C11-hydrocarbons), demonstrating a different type of unsaturation that is active at picomolar concentrations [24].

The chemical composition of algae depends on the growth environment and season, and when dry, they can be used as feed in the rations of ruminants. Choi et al. [25] analyzed residues of Sargassum fusiforme, and they reported that dried algae could contain up to 30% crude protein, indicating that some algae can be considered protein supplements. This contrasts with our results obtained using Kelp, Ulva, and Silk, which achieved CP percentages of 10.4, 10.5, and 6.6 % of CP, respectively, consistent with other reports that have mentioned that the protein content depends on the algae type: brown (4–24% CP of DM), green (9–33% CP of DM), or red (8–47% CP of DM) [26]. Furthermore, the process or protocol used to evaluate protein levels can affect the quantification precision in either the total nitrogen or true protein. A traditional conversion of nitrogen to protein (factor of 6.25) may result in an overestimation of the protein content of seaweed [27,28]. The ether extract (total lipids) was reported to range from 0.2% to 8% [29] for green algae, which is consistent

with what we found for Ulva, Kelp, and Silk. However, the total lipids in the seaweed's content of polyunsaturated fatty acids needs to be greater for use with terrestrial crops [30]. The oil from algae is used in foods and medicine. At the same time, differences in the lipid and fatty acid contents were attributed to species and environmental conditions. In all cases, the lipid content was lower than 5 %, with most species presenting lipid contents lower than or around 1 % [31]. A high content of structural carbohydrates and a variable content of ash is a characteristic of the chemical composition of macroalgae [32]; this has awakened an interest in industrial exploitation, especially in the sense of the compounds related to polysaccharide fraction ranging from 4 to 76% DM [33]. It is important to mention that fibrous carbohydrates (NDF) ranged from 8 to 58% in dry macroalgae [34], which is consistent with our results. However, the industry is focused on other polysaccharides, such as alginate, laminarin, and fucoidan in brown algae and agar and carrageenan in red algae [31]. Ash content is another compound that is highly variable in quantity according to the algae species, environment, season, and region [34]. According to Pardilhó et al. [31], the amount of ash can range from 7 to 50%, with Ulva being the most generous despite presenting the lowest content; in contrast, we found the Silk has the lowest amount of ash (20.9%). However, seaweeds may contain concentrations of heavy metals and minerals from seawater and may contain several times the ash content of land plants [32], limiting their gross energy value and requiring regular monitoring [30]. Thus, the three algae evaluated in the current study should be used in a total mixed ration (TMR) to show the benefits demonstrated in our research, such as its use as a protein source.

Numerous studies were conducted to determine the in situ or in vivo ruminal degradability of different algae. However, the results present high variation depending on the animal species, type of algae, and technique used. Ulva algae presented 38% effective rumen degradation in Holstein cows but 47% effective rumen degradation in goats [35,36]. For Laminaria, 72% DM degradability was observed after 48 h when incubated in sheep rumen [37]; this could be explained by the passage rate and specific characteristics of the ruminal process of each species. For Ulva, in vitro DM degradability of 70% was reached at 72 h, which is superior to what has been reported in vivo or in situ, but Zitouni et al. [36] mentioned that in vitro techniques generally provide greater degradability values than in situ ones. For Kelp, 76% DM degradability has been observed when using a TMR of 30% [38]. These results are higher than those that we obtained for Kelp algae (61.5 % DM in vitro degradability) but are in concordance with Casas et al. [39], who found in situ digestibility values of 71.4%. The different among that study and ours is that they added algae as a component of the total diet, and we only evaluated the algae without using it as a dietary addition.

The increased digestibility in some species of algae when used as an additional supplement could be attributed to their alginate concentration and the NDF:ADF ratio [40]. In vitro gas production is a good indicator of ruminal fermentation patterns [41]. Greater amounts of nutrients in the rations indicate a better nutrient availability for rumen microorganisms [42], stimulating the degradability of different nutrients [43]. Zitouni et al. [36] reported 42 mL/g OM of gas production for Ulva after 24 h, which is similar to what we reported (44.37 mL/g DM) after 48 h Likewise, Ulva presents a high mineral content and low crude protein, but these could contribute to biomass production rather than gas production. Dubois et al. [34] mentioned that an elevated protein content in algae results in greater in vitro gas production. Likewise, some algae that contain polysaccharides, carrageenans, alginate, fucoidans, agar, ulvans, xylans, laminarin, and florideans starch [44], limit the availability of nutrients to rumen microbiota. Brown algae produce phlorotannin, which is a polar, non-phenolic metabolite that acts as a defense mechanism [45], and when algae are consumed by ruminants, it could alter rumen microbial communities. The number of protozoa (related to rumen methane production) declines after incubation in the presence of this compound [46]. In this context, Belanche et al. [47] reported that phlorotannin has potential anti-protozoal properties and that it can reduce total gas production.

Ruminal fermentation produces gases. The main compounds are hydrogen, $CO_2$, and $CH_4$. The algae's nutritional value is insufficient to present high fermentation, and therefore, it does not produce large amounts of $CO_2$ and $CH_4$. Algae can contain large amounts of ether extracts and can have high eicosapentaenoic and docosahexaenoic (DHA) contents in particular. These long-chain fatty acids are related to reduced $CH_4$ production [48]. Some authors have reported a reduction in $CH_4$ of 80% when using a DHA-rich supplement. Lee-Rangel et al. [49] mentioned that using saturated fatty acids could be an efficient strategy to mitigate methane in vitro ruminal fermentation. Nevertheless, there is evidence to suggest that the secondary metabolites of algae function to decrease ruminal $CH_4$ production during enteric fermentation.

Current studies suggest that some seaweeds contain halogenated compounds that have the potential to reduce $CH_4$ production [50]. Another type of compound that could have the same effect is phlorotannins, but the authors report that they can have a positive or negative impact on rumen function and $CH_4$ production [51]. Furthermore, studies have demonstrated bromoform (halogenated compound) supplementation or supplementation with seaweed that contains this can reduce $CH_4$ production (50% to 95%) and inhibit methanogenesis without negative effects on ruminal fermentation or animal growth performance [52]. However, if the ruminal medium is influenced, the risk of decreased DMI [53] affects ruminal degradability [54] and may alter ruminal microbiota [53], and growth and productivity can be affected. It is important to consider that some of these compounds are present in fresh form, but seaweed generally is treated to preserve it. The researchers did not consider post-harvest processing methods, such as freezing or drying.

## 5. Conclusions

Due to their nutritional contents, three types of algae—Kelp (*Macrocystis pyrifera*), Ulva (*Ulva* spp.), and Silk (*Mazzaella* spp.)—can be used as food additives in ruminant feed in livestock systems. Our study proposes the use of these three types of algae in a total mixed ration to potentialize the benefits of numerous bioactive compounds that could benefit ruminant health or livestock production. Our study suggests that the seaweeds reported in the current research require further investigation to describe their fundamental role as a source of positive bioactive compounds in a total mixed ration and to characterize the bioactive compounds to determine that they remain at least partially unaffected in the ruminant metabolism so that they can be transferred to milk or meat depending on the livestock systems.

**Author Contributions:** Conceptualization, H.A.L.-R. and J.A.R.-J.; methodology, H.A.L.-R.; software, H.A.L.-R.; validation, H.A.L.-R., G.Á.-F. and A.J.C.-C.; formal analysis, H.A.L.-R.; investigation, H.A.L.-R. and G.Á.-F.; resources, H.A.L.-R. and A.J.C.-C.; data curation, H.A.L.-R.; writing—original draft preparation, H.A.L.-R. and J.A.R.-J.; writing—review and editing, H.A.L.-R. and J.A.R.-J.; visualization, R.O.C.-L. and J.I.A.-V.; supervision, J.A.M.-G.; project administration, H.A.L.-R. and A.J.C.-C.; funding acquisition, H.A.L.-R. and A.D.l.C.-G. All authors have read and agreed to the published version of the manuscript.

**Funding:** This research received no external funding.

**Institutional Review Board Statement:** The sampling procedures in the sheep flock were reviewed and approved by the Committee for the Ethical Use of Animals in Experiments of the Universidad Autonoma Metropolitana and Universidad Autonoma de San Luis Potosi according to the regulations and standards that the Mexican government requires for the use of animals for several diverse activities (BP-PA-20210513092803518-1069570).

**Informed Consent Statement:** Not applicable.

**Data Availability Statement:** The data presented in this study are available from the corresponding author upon request. The data are not publicly available due to institutional instructions.

**Conflicts of Interest:** The authors declare no conflict of interest.

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
