# Peer review of "Evaluation of Three Marine Algae on Degradability, In Vitro Gas Production, and CH4 and CO2 Emissions by Ruminants"

_fermentation, doi:10.3390/fermentation8100511_

Round 1

Reviewer 1 Report

Dear authors. This is an interesting study, but I have some issues with the study design and presentation and interpretation of the results, which you may address during revision of your manuscript.

Please carefully revise the manuscript in terms of correct use of English language, because it is often hard to read and to understand what you want to say. I recommend you to use a professional editing service.

Please find specific comments in the following:

Abstract: L17-18, which alternatives? an introducing sentence is missing here; L25-26, this sentence is redundant; please add to the abstract some notes on study design, the used in vitro system, substrates or rations (experimental units), and, implications for practice.

Introduction: L47, better "exposed" than "produced", the microbes produce methane; L51, bacteria or archaea, word is missing; L53, Besides . . . word is missing; L60, ". . . being used as livestock", . . . as additive in feeding livestock?

Materials and methods: L83, please explain the sentence; L111, FDN and FDA, why do you not use the common abbreviations aNDFom and ADFom? L125-126, you used a vacuum pump to obtain ruminal fluid? please explain that; add information on pH, redox potential and temperature of the ruminal fluid and on no. of biological and technical replicates; which substrates, which rations, which in vitro system did you use for the incubations? L137-140, better describe the procedure, give a reference, what was your starting value, did you account for microbial biomass production, any correction applied? L145, what diets? you only incubated the algae alone, right? why not combinations or TMR with several substitution levels? the animal will probably never receive a pure algae diet . . . in vitro methods offer the opportunities to investigate much proper variants, although I know that they can easily become very labourious; L153, "Statistical" not "Statical"; please provide more detail on the statistical tests that were used.

Results: L164-166, delete; please check order of tables as you start with Table 2; Table 2, mention the database here; give a short explanation what the relevance index describes - not everybody is familiar with that; L179-180, I think EE is negligible, but you should outline the very high ash contents; Table 1, are the DM concentrations that of pre-dried material? you should give DM concentration of the original material, please add energy concentration, if possible, at GE, DE, ME, and NEL level (see e.g., Menke and Steingass, ref. 14); it would be very helpful for the reader if you could keep colour for the algae variants constant among figures; Fig. 2, this figure shows gas production not DM degradation as stated; Table 3, the unit of gas production rate should be mL/h, why did you add methane and carbon dioxide concentration here when this is given in Fig. 3? the results of DM degradation are not shown.

Discussion: imply what these nutrient profiles allow . . . how much (to which level) can they be used in a ration? discuss the high ash concentrations; can the algae be fed without a ration? do the animals eat it - what is about odor and taste? are there some information on antinutritional compounds or adverse effects? discussion on OM or protein degradation/digestibility would be of greater interest; L273, again, you will surely not feed the algae without a TMR; L274-275, . . . and OM concentration, and nutrient concentrations; L276, "Greater" instead of "Grater".

Conclusions: L317-318, this suggestion based on which data?

Author Response

Reviewer #1

R1: Dear authors. This is an interesting study, but I have some issues with the study design and presentation and interpretation of the results, which you may address during revision of your manuscript.

AR: We appreciate your comments and recommendations for improving the manuscript's quality. The changes were highlighted using the tracking changes to mark the editions on the manuscript.

R1: Please carefully revise the manuscript in terms of correct use of English language, because it is often hard to read and to understand what you want to say. I recommend you to use a professional editing service.

AR: We appreciate the comment. The manuscript was sent to the English language edition to improve the quality of the current manuscript.

R1: Please find specific comments in the following:

Abstract: L17-18, which alternatives? an introducing sentence is missing here;

AR: We appreciate the comment. The concern was addressed, and the lines changed.

L25-26, this sentence is redundant; please add to the abstract some notes on study design, the used in vitro system, substrates or rations (experimental units), and, implications for practice.

AR: We appreciate the comment. The concern was addressed, and the lines changed.

Introduction

L47, better "exposed" than "produced", the microbes produce methane;

AR: We appreciate the comment. The concern was addressed, and the lines changed.

L51, bacteria or archaea, word is missing;

AR: We appreciate the comment. The concern was addressed, and the lines changed.

L53, Besides . . . word is missing;

AR: We appreciate the comment. The concern was addressed, and the lines changed.

 L60, ". . . being used as livestock", . . . as additive in feeding livestock?

AR: We appreciate the comment. The concern was addressed, and the lines were edited.

Materials and methods:

L83, please explain the sentence;

AR: We appreciate the comment. The concern was addressed, and the lines were edited.

L111, FDN and FDA, why do you not use the common abbreviations aNDFom and ADFom?

AR: We understand the reviewer's concern. However, according to the editorial of animal feed science and technology (2005), and recent studies, we evaluate neutral detergent fiber (NDF) and acid detergent fiber (ADF). The difference with the reviewer's comment is that a NDFom and NDFom are expressed exclusive of residual ash, and in contrast, we expressed the values of fibre, including the residual ash. Thus, we decided to use the corresponding abbreviations when we wrote the current manuscript.

https://doi.org/10.1016/j.anifeedsci.2004.11.011

 L125-126, you used a vacuum pump to obtain ruminal fluid? please explain that; add information on pH, redox potential and temperature of the ruminal fluid and on no of biological and technical replicates; which substrates, which rations, which in vitro system did you use for the incubations?

AR: We appreciate the comment. The concern was addressed, and the lines were edited.

L137-140, better describe the procedure, give a reference, what was your starting value, did you account for microbial biomass production, any correction applied?

AR: We appreciate the comment. The concern was addressed, and the lines were edited.

L145, what diets? you only incubated the algae alone, right? why not combinations or TMR with several substitution levels? the animal will probably never receive a pure algae diet . . . in vitro methods offer the opportunities to investigate much proper variants, although I know that they can easily become very labourious;

AR: We understand the concern of the reviewer. The description by the reviewer is correct. However, this is our first study that aimed to describe the applicability of the three algae in ruminant feed. Thus, we obtained valuable information that will help us in future research.

Currently, we are testing the inclusion of the three types of algae in lamb diets (TMR) to describe in a study in vivo how the algae benefit the feeding of ruminants, metabolism, and meat metabolomic quality. We hope to send these results in a short time as the second chapter of this project.

L153, "Statistical" not "Statical"; please provide more detail on the statistical tests that were used.

AR: We appreciate the comment. The concern was addressed, and the lines were edited.

Results:

L164-166, delete; please check order of tables as you start with Table 2; Table 2, mention the database here; give a short explanation what the relevance index describes - not everybody is familiar with that;

AR: We appreciate the comment. The concern was addressed, and we edited the lines to better descriptions to help the readers to understand the information.

L179-180, I think EE is negligible, but you should outline the very high ash contents; Table 1, are the DM concentrations that of pre-dried material? you should give DM concentration of the original material, please add energy concentration, if possible, at GE, DE, ME, and NEL level (see e.g., Menke and Steingass, ref. 14); it would be very helpful for the reader if you could keep colour for the algae variants constant among figures;

AR: We understand the concern of the reviewer. First, we added a better description in table 2. The chemical composition is % DM basis. In table 3, we included the complete information based on the model by Menke and Steingass. Also, we re-edited the section for a better description and included the calculations in materials and methods. In figures 1, 2, and 3, we standardized the colors for algae description. We want to be thankful to the reviewer for these suggestions and comments.

Fig. 2, this figure shows gas production not DM degradation as stated;

AR: We appreciate the comment. The concern was addressed, and the lines changed.

Table 3, the unit of gas production rate should be mL/h, why did you add methane and carbon dioxide concentration here when this is given in Fig. 3? the results of DM degradation are not shown.

AR: The concern was addressed and changed. The reviewer is correct. We re-edited the table.

Discussion:

imply what these nutrient profiles allow . . . how much (to which level) can they be used in a ration? discuss the high ash concentrations; can the algae be fed without a ration? do the animals eat it - what is about odor and taste? are there some information on antinutritional compounds or adverse effects? discussion on OM or protein degradation/digestibility would be of greater interest;

AR: The concern was addressed and changed. The reviewer is correct. We re-edited the discussion section.

L273, again, you will surely not feed the algae without a TMR;

AR: As we mentioned before, we appreciate the comment and the observation by the reviewer. This is our first study that aimed to describe the applicability of the three algae in ruminant feed. Currently, we are testing the inclusion of the three types of algae in lamb diets (TMR) to describe a study in vivo.

L274-275, . . . and OM concentration, and nutrient concentrations;

AR: The concern was addressed and changed.

L276, "Greater" instead of "Grater".

AR: The concern was addressed and changed.

Conclusions: L317-318, this suggestion based on which data?

AR: The concern was addressed and changed. We re-edited the conclusion section.

Reviewer 2 Report

The title of this article is:

The degradability, in vitro gas production, and CO2 and CH4 emissions by ruminants of three marine algae were evaluated

 I have read with great interest the research article that was submitted by the authors on the topic of greenhouse emissions and how to prevent them with different algae. Currently, I think it is a necessity in order to reduce the greenhouse gas emissions in order to protect the environment. However, it requires a lot of improvements in terms of the methods, the interpretation of the data, etc.

 For their experiments, the authors have used animals to collect fluid from the rumens of the animals. However, I have been unable to find any ethical statements. I would appreciate it if you could clarify this.

 In order to identify the name of the algae samples, how did the authors identify them? For algae samples, it is necessary to include the place of collection as well as the source of the sample.

 The main focus of this research is on reducing greenhouse gas emissions. A number of bacterial groups are responsible for the production of methane and carbon dioxide in general. It would be appreciated if authors could analyze microbial changes in rumen samples before/after experiments were conducted, as it would be helpful for future research.  

 There is a lack of adequate data presentation and interpretation in the results section.

 All experimental data should be subjected to statistical analysis.

 Throughout the manuscript, there are a number of incomplete sentences.

 There is a need to improve the language of the manuscript.

Author Response

Reviewer 2

The title of this article is:

The degradability, in vitro gas production, and CO2 and CH4 emissions by ruminants of three marine algae were evaluated

 I have read with great interest the research article that was submitted by the authors on the topic of greenhouse emissions and how to prevent them with different algae. Currently, I think it is a necessity in order to reduce the greenhouse gas emissions in order to protect the environment. However, it requires a lot of improvements in terms of the methods, the interpretation of the data, etc.

AR: We appreciate your comments and recommendations for improving the manuscript's quality. The changes were highlighted using the tracking changes to mark the editions on the manuscript.

 For their experiments, the authors have used animals to collect fluid from the rumens of the animals. However, I have been unable to find any ethical statements. I would appreciate it if you could clarify this.

AR: We appreciate your comments and recommendations for improving the manuscript's comprehension to the readers. We added information about the ethical statements in the materials and methods section. Also, we describe extensively the Institutional Review Board Statement.

 In order to identify the name of the algae samples, how did the authors identify them? For algae samples, it is necessary to include the place of collection as well as the source of the sample.

AR: We appreciate your comments and recommendations for improving the manuscript's comprehension to the readers. We added information about the algae harvesting.

 The main focus of this research is on reducing greenhouse gas emissions. A number of bacterial groups are responsible for the production of methane and carbon dioxide in general. It would be appreciated if authors could analyze microbial changes in rumen samples before/after experiments were conducted, as it would be helpful for future research.  

AR: We understand the concern of the reviewer. The description by the reviewer is correct. As we mentioned before to reviewer number 1, this is our first study that aimed to describe the applicability of the three algae in ruminant feed. Thus, we obtained valuable information that will help us in future research.

Currently, we are testing the inclusion of the three types of algae in lamb diets (TMR) to describe in a study in vivo how the algae benefit the feeding of ruminants, metabolism, and meat metabolomic quality. One of our exciting points is the description of the microbiome using a metagenomic analysis to understand how the VOCs modify the bacteria from the algae.

We hope to send these results in a short time as the second chapter of this project.

 There is a lack of adequate data presentation and interpretation in the results section.

AR: We appreciate the comment. The concern was addressed, and we edited the lines to better descriptions to help the readers to understand the information.

 All experimental data should be subjected to statistical analysis.

AR: We understand the reviewer’s concern. Only the chemical composition was not statistically analyzed. However, we considered that the algae were harvested simultaneously and in the zone. Thus, more amounts of algae and at different times of harvesting to make robust statistics

Throughout the manuscript, there are a number of incomplete sentences.

AR: We appreciate the comment. The concern was addressed, the manuscript was edited and later we sent it to the language edition in the MDPI system.

There is a need to improve the language of the manuscript.

AR: We appreciate the comment. The manuscript was sent to the language edition.

Round 2

Reviewer 1 Report

The revisions the authors made are convincing. You should add units to model parameters described in lines 155-160. The units given in Table 3 are probably not correct: V is mL and s is mL/h. Abbreviation for lag phase is L not l.

Author Response

R1: The revisions the authors made are convincing. You should add units to model parameters described in lines 155-160. The units given in Table 3 are probably not correct: V is mL and s is mL/h. Abbreviation for lag phase is L not l.

AR: We appreciate your comments and recommendations for improving the manuscript's quality. The changes were highlighted using the tracking changes to mark the editions on the manuscript. We added units to model parameters described in lines 155 to 160. Also, the units described in Table 3 were modified.

Reviewer 2 Report

I have reviewed the entire manuscript in detail. In response to all questions and doubts raised by the reviewer, the authors have provided adequate responses. The authors have also improved the quality of the manuscript over the earlier submission. In the present format, I believe the paper is suitable for acceptance.

Author Response

R2: I have reviewed the entire manuscript in detail. In response to all questions and doubts raised by the reviewer, the authors have provided adequate responses. The authors have also improved the quality of the manuscript over the earlier submission. In the present format, I believe the paper is suitable for acceptance.

AR: We appreciate your comments and recommendations for improving the manuscript's quality.